# Multi-Dimensional Factors Associated with Illegal Substance Use Among Gay and Bisexual Men in Taiwan

**DOI:** 10.3390/ijerph16224476

**Published:** 2019-11-14

**Authors:** Dian-Jeng Li, Shiou-Lan Chen, Cheng-Fang Yen

**Affiliations:** 1Graduate Institute of Medicine, College of Medicine, Kaohsiung Medical University, Kaohsiung 80708, Taiwan; u108800004@kmu.edu.tw (D.-J.L.); shioulan@kmu.edu.tw (S.-L.C.); 2Department of Addiction Science, Kaohsiung Municipal Kai-Syuan Psychiatric Hospital 80276, Kaohsiung 80708, Taiwan; 3Department of Psychiatry, Kaohsiung Medical University Hospital, Kaohsiung 80708, Taiwan

**Keywords:** sexual minorities, illegal substance use, homophobic bullying

## Abstract

Illegal substance use in sexual minorities is an important health issue worldwide. The present cross-sectional study aimed to investigate the multi-dimensional factors associated with illegal substance use among gay and bisexual men in Taiwan. This questionnaire-survey study recruited 500 gay or bisexual men aged between 20 and 25 years. Their experiences of using eight kinds of illegal substances in the preceding month were collected. Their previous experiences of homophobic bullying, satisfaction with academic performance, truancy, perceived family and peer support in childhood and adolescence, and social-demographic characteristics, were also collected. Potential factors associated with illegal substance use were identified using univariate logistic regression, and further selected into a forward stepwise logistic regression model to identify the factors most significantly related to illegal substance use. A total of 22 (4.4%) participants reported illegal substance use in the preceding month, and mean age was 22.9 ± 1.6. Forward stepwise logistic regression revealed that being victims of homophobic cyberbullying in childhood and adolescence (odds ratio (OR) = 1.26; *p* = 0.011), disclosure of sexual orientation at junior high school (OR = 4.67; *p* = 0.001), and missing classes or truancy in senior high school (OR = 2.52; *p* = 0.041) were significantly associated with illegal substance use in early adulthood. Multi-dimensional factors in childhood and adolescence that were significantly associated with illegal substance use in early adulthood among gay and bisexual men were identified. Besides traditional bullying, the effect of cyberbullying and school performance on illegal substance use should not be ignored. This study is limited to the cross-sectional design and possible recall bias. Mental health professionals must routinely assess these significant factors to prevent and intervene in illegal substance use among gay and bisexual men.

## 1. Introduction

### 1.1. Substance Use in Sexual Minorities

Substance use has become a major public health concern. According to the 2018 World Drug Report, 2.75 hundred million people have used illegal drugs at least once, which comprises 5.6% of the general population aged between 15 and 64 years [1]. Among them, 31 million people have been diagnosed with substance use disorder. Substance use often results in socio-economical and health burdens, including domestic violence [2,3], increased crime rates [4], suicide [5], comorbidity with mental illnesses [6], and comorbidity with physical illnesses, such as blood-borne disease [7]. Substance abuse is also a key public health issue worldwide for sexual minorities. An epidemiologic study derived from the National Epidemiologic Survey on Alcohol and Related Conditions-III, reported a higher prevalence of drug abuse at some point during their lifetime, for gay/lesbian (19.6%) and bisexual (26.5%) individuals, compared with heterosexual individuals (12.1%) [8]. In Taiwan, a cross-sectional study indicated 16% of recreational drug use in the previous 6 months within men who have sex with men (MSM; 98.6% of them are gay/bisexual men) [9], and it is also higher than an epidemiological study, which reported 0.17% of past-1-year prevalence for club drug use in general population [10]. To be specific, methamphetamine along with ketamine and marijuana are popular according to a national survey [10]; however, such investigation for sexual minorities is insufficient. A previous study indicated that poly drug use among gay and bisexual men, is significantly associated with HIV infection, high-risk sexual practices, and partner violence [11]. Meyer proposed the minority stress theory (MST) to illustrate the association between multiple stressors and mental health status in sexual minorities [12]. This model was tested for substance use and it showed a good level of application in sexual minorities [13]. Investigation of the factors affecting substance use in sexual minorities can serve as the basis for developing prevention and intervention programs.

### 1.2. Factors Related to Substance Use in Sexual Minorities

According to the ecological systems theory developed by Bronfenbrenner [14], substance use in sexual minority individuals is an ecological phenomenon, which has been established and perpetuated over time because of the complex interactions between individual and social factors. With regards to individual factors associated with substance use, previous studies have indicated that bisexual individuals have a higher rate of substance use compared with gay/lesbian and heterosexual individuals [15,16]. Moreover, some studies have found that disclosure of sexual orientation was associated with significantly greater substance use [17,18,19], while other studies do not support this finding [20,21]. To the best of our knowledge, whether disclosure of sexual orientation at a specific life stage is associated with substance use in sexual minority individuals has not been previously examined. Given that different individuals may have various abilities and strategies to cope with stress at various life stages [22,23], and that substance use is one common but improper coping strategy [24,25], it is possible that disclosure of sexual orientation at a specific life stage may increase the risk of substance use in sexual minority individuals.

With regard to social factors associated with substance use, social support from peers and family contributes to the mental health of sexual minority individuals [26,27]. Illegal substance use in sexual minority individuals has been previously reported to be associated with family rejection [28]. Individuals who perceived high levels of peer support were more likely to use legal substances, such as cigarettes, compared with those who perceived low levels of peer support [29,30,31]. In addition to family and peers, schools are also an important social environment that can affect the growth of most individuals. Research has revealed that low academic performance in school is associated with substance use and emotional distress in sexual minority individuals [32]. Moreover, missing classes, truancy, and substance use are presentations of a poor adjustment to school life [33]. However, no study has taken the association between illegal substance use and multiple social factors, including social family support, peer support, satisfaction with academic performance, missing classes, and truancy, into consideration in sexual minority individual.

### 1.3. Association between Homophobic Bullying and Substance Abuse in Sexual Minority Individuals

Homophobic bullying is also a noteworthy issue for sexual minorities. Homophobia is defined as negative beliefs, attitudes, and behaviors toward Lesbian, gay, bisexual and transgender (LGBT) individuals [34]. A meta-analytic study revealed higher rates of depression and suicide in LGBT individuals [35]. Mental health problems are reported to be mainly a result of negative life experiences, including homophobic bullying [36]. Sexual minority individuals may encounter stress associated with unfriendly social environments full of prejudice and discrimination, which may lead to mental health problems [37,38]. Furthermore, homophobic bullying in childhood and adolescence has been reported to be harmful to future psychosocial and health outcomes in adulthood [39]. Therefore, investigation into the effects of homophobic bullying can help clinicians with early intervention.

The results of previous studies on the association between homophobic bullying and substance abuse in sexual minority individuals are mixed. Although a recent study found that homophobic bullying and victimization were not significantly associated with current substance use among sexual minorities [40], another study demonstrated that lesbian, gay, and bisexual youths reporting a high level of at-school victimization reported more severe substance use than those experiencing a low level or no victimization [41]. Moreover, previous studies mainly examined the role of traditional bullying victimization for substance abuse, whereas the role of cyberbullying victimization, a new type of harassment that has emerged in the digital age [42], in substance abuse warrants further study. It has been previously reported that sexual minority youths experience higher online peer victimization compared with heterosexual youths [43]. The potential association between both traditional homophobic bullying and cyberbullying with illegal substance use in sexual minority individuals, warrants further investigation.

### 1.4. Aims of the Study

The aim of the present cross-sectional study was to investigate multi-dimensional factors of illegal substance use in gay and bisexual men in Taiwan. The authors suppose that there are multi-dimensional factors, including homophobic bullying victimization, sexual orientation characteristics, and family, peer, and school factors that are associated with illegal substance use in early adulthood among gay and bisexual men.

## 2. Materials and Methods

### 2.1. Participants

In the current study, we recruited participants using an online advertisement that was posted on a bulletin board system, Facebook, and the home pages of five health promotion and counseling centers for sexual minorities in Taiwan. Print versions of the advertisement were mailed to the LGBT student clubs at 25 colleges in Taiwan. Those who exhibited any cognitive impairment (e.g., substance intoxication or intellectual disability) that prevented them from understanding the goal of the study or from completing the questionnaires were not included in this study. In total, 500 gay or bisexual men aged between 20 and 25 years were recruited into this study. Before assessment, informed consent was obtained from all participants. Before the recruitment, the study was also approved by the Institutional Review Board of Kaohsiung Medical University Hospital (KMUHIRB-F(I)-20150026).

### 2.2. Measures

#### 2.2.1. Illegal Substance Use

The D-score of Drug Use Disorders Identification Test-Extended (DUDIT-E) was used to identify the history of illegal substance use for all individuals. It had been developed for the sequential clinical assessment of drug use. The concurrent validity of the D-score is reported to be acceptable, and test-retest reliability is 0.79, indicating an excellent intraclass correlation [44]. The participants were asked about any previous experiences where they used marijuana, methamphetamine, cocaine, heroin, ketamine, ecstasy, hallucinogens, volatile organic compounds, and other drugs in the preceding month. Participants who had used any kind of illegal substance were classified as having illegal substance use.

#### 2.2.2. Experiences of Homophobic Bullying and Victimization

Six items from the Chinese version of the self-report School Bullying Experience Questionnaire (C-SBEQ) [45] were used to assess the experiences of the participants with regard to traditional bullying in primary (grades 1–6), junior high (grades 7–9), and senior high (grades 10–12) schools, based on their gender role nonconformity and sexual orientation at school, tutoring schools, after-school classes, and part-time workplaces. Multiple forms of traditional homophobic bullying victimization were evaluated, including name calling, social exclusion, ill speaking, forced work, physical abuse, and confiscation of money, school supplies, and snacks. The responses for these six items were graded on a 4-point Likert scale as following: 0 = never, 1 = just a little, 2 = often, and 3 = all the time. A previous investigation on C-SBEQ psychometrics revealed that the C-SBEQ has acceptable validity and reliability [45]. The Cronbach α value of the scale for evaluating the homophobic bullying victimization was 0.82. In this study, participants who rated 2 to 3 for any item were identified as self-reported victims of homophobic traditional bullying. In order to group the subtype of traditional bullying, we categorized social exclusion into “social bullying”, name calling along with ill speaking into “verbal bullying”, physical along with forced work into “physical abuse”, and confiscation of money/supplies into “snatching of belongings”.

Three items from the Cyberbullying Experiences Questionnaire [46] were also used to evaluate the participants’ cyber bullying experiences in elementary, junior high, and senior high schools based on their gender role nonconformity and sexual orientation. These three items described the experiences of others posting mean or hurtful comments, others posting upsetting photos, pictures, or videos, and online rumor-spreading through blogs, emails, social media platforms (Facebook, Twitter, Plurk), and pictures or videos. An example of the questions is as follows: “How often have other students posted mean or hurtful comments about you through emails, blogs, or social media because they thought of you as a sissy (they found you homosexual or bisexual)?” The responses to these items were graded on a 4-point Likert scale, with scores ranging from 0 (never) to 3 (all the time). The Cronbach α value of the scales for evaluating homophobic cyberbullying victimization was 0.81. In this study, participants who rated 1 for any item were marked as self-reported victims of homophobic cyberbullying.

#### 2.2.3. Demographic and Family Characteristics

Data were recorded on the participants’ age, educational level, parental marriage status, and paternal and maternal education levels. We divided included participants were into those with high education levels (college or above) and those with low education levels (high school or below). We also grouped participants into those with high parental education levels (parents completed 9 years of compulsory fundamental education) and those with low parental education levels (parents did not complete 9 years of compulsory fundamental education).

#### 2.2.4. Sexual Orientation Characteristics

The sexual orientation of participants (homosexual or bisexual) and disclosure of sexual orientation to others at primary school, junior high school, senior high school, and college, was recorded.

#### 2.2.5. Social Support

The Chinese version of the 5-item self-administered Family Adaptation, Partnership, Growth, Affection, Resolve (APGAR) Index was used to estimate participants’ satisfaction with different domains of family support during their childhood and adolescence [47,48]. Each item was rated on a 4-point Likert scale, with scores ranging from 0 (never) to 3 (always). The Family APGAR Index was also transformed into the Peer APGAR Index to identify the participants’ satisfaction with different domains of their peer support during childhood and adolescence. Higher total scores on the Family and Peer APGAR indices stood for higher levels of family and peer support, respectively. The Cronbach α values for the Family and Peer APGAR indices in the present study were 0.86 and 0.87, respectively.

#### 2.2.6. School Characteristics

We invited participants to retrospectively remind their subjective satisfaction with their academic performance in primary, junior high, and senior high schools using the 4-point Likert scale, ranging from 0 (very satisfied) to 3 (not satisfied at all). In this study, participants who answered 2 or 3 were identified as dissatisfied with their academic performance. The tendency to miss classes or truancy in elementary, junior high, and senior high schools was evaluated using a 4-point Likert scale, ranging from 0 (never) to 3 (very frequent). In this study, all participants who did not answer 0 on any item were classified as having a tendency to miss classes or truancy.

### 2.3. Procedure

This study was performed as a paper-and-pencil questionnaire. Assistants in our research team individually explained the procedures and methods for completing the research questionnaires to the participants. The participants could ask any question when they faced difficulties in completing the questionnaires, and the research assistants would answer them.

### 2.4. Statistical Analysis

Initially, we summarized the demographic, sexual orientation, family, school, and social support characteristics in Table 1. In order to estimate the odds ratio (OR) for multiple variables, univariate logistic regression was used to identify potential predictors associated with illegal substance use. Second, all potential predictive variables (*p* < 0.05) identified from the first step were selected in a forward stepwise logistic regression model to determine the best predictors for illegal substance use. All tests were 2-tailed, and statistical significance was set at *p* < 0.05. All data were processed using SPSS version 23.0 for Windows (SPSS Inc., Chicago, IL, USA).

## 3. Results

### 3.1. Patient Variables

A total of 500 males, 371 gay men and 129 bisexual men participated in the current study. The mean age of the participants was 22.94 ± 1.57, and 22 (4.4%) men reported using illegal substances in the preceding month. Illegal substance use by the participants included cannabis (*n* = 4; 0.8%), methamphetamines (*n* = 7; 1.4%), ketamine (*n* = 1; 0.2%), inhalants (*n* = 1; 0.2%), and poly-substance use (*n* = 9; 1.8%). The frequency of illegal substance use was as follows: marijuana (never used: 99%; ever used but less than once per month: 1%), methamphetamine (never used: 97%; ever used but less than once per month: 1.8%; once per month but less than twice per month: 0.2%; twice to four times per month: 1%), ketamine (never used: 98.6%; ever used but less than once per month: 1.2%; once per month but less than twice per month: 0.2%), ecstasy (never used: 98.8%; ever used but less than once per month: 1%; once per month but less than twice per month: 0.2%), volatile organic compounds (never used: 99.6%; ever used but less than once per month: 0.2%; twice to thrice per week: 0.2%). The frequency of bullying was as follows: traditional bullying (never: 13%; just a little: 49%; often/all the time: 38%) and cyberbullying (never: 59.8%; just a little: 32.6%; often/all the time: 7.6%). The demographic and sexual orientation characteristics of all individuals are listed in Table 1. The family and school characteristics and social support findings are listed in Table 2.

### 3.2. Predictors of Illegal Substance Use

The results of the univariate logistic regression analysis showed that victims of homophobic traditional bullying in school (OR = 3.00; *p* = 0.015), victims of homophobic cyberbullying in school (OR = 4.22; *p* = 0.003), disclosure of sexual orientation at junior high school (OR = 5.16; *p* < 0.001), lower education level (OR = 3.70; *p* = 0.009), and missing classes or truancy in senior high school (OR = 2.68; *p* = 0.025) were significantly associated with the use of illegal substances (Table 1 and Table 2). Moreover, the association between subtypes of traditional bullying and specific illegal substance were also identified by univariate logistic regression. We found that social bullying was significantly associated with use of methamphetamine (OR = 5.52; *p* = 0.028). On the other hand, there was an insignificant trend of association between use of marijuana versus snatching of belongings (OR = 10.00; *p* = 0.051), and use of methamphetamine versus verbal bullying (OR = 4.79; *p* = 0.063).

The results of the forward stepwise logistic regression revealed that victims of homophobic cyberbullying in school (OR = 1.26; *p* = 0.011), disclosure of sexual orientation at junior high school (OR = 4.67; *p* = 0.001), and missing classes or truancy in senior high school (OR = 2.52; *p* = 0.041) were significantly associated with the use of illegal substances (Table 3).

## 4. Discussion

The present study reports that 4.4% of gay and bisexual men used illegal substances in the preceding month, and that multi-dimensional factors were associated with illegal substance use among gay and bisexual men, including homophobic cyberbullying victimization in childhood and adolescence, disclosure of sexual orientation at junior high school, and missing classes or truancy in senior high school.

### 4.1. Rate of Illegal Substance Use in Sexual Minorities

The reported prevalence of illegal substance use in gay and bisexual men varies depending on the participants’ age and periods of substance use surveyed. For example, a national survey in the United States showed a lifetime prevalence of substance-dependence of 5.7% for gay and bisexual men [49]. Whereas O’Cleirigh and colleagues reported a drug use rate of 53.1% among HIV-infected gay and bisexual men over the past three months [50]. The present study focused on a group of gay and bisexual men aged between 20 and 25 years. The rate of illegal substance use among gay and bisexual men of different ages in Taiwan warrants further study.

### 4.2. Homophobic Traditional and Cyber Victimization and Illegal Substance Use in Sexual Minorities

The results of the univariate logistic regression in the present study revealed that homophobic traditional victimization in childhood and adolescence was significantly associated with illegal substance use in early adulthood among gay and bisexual men. Sexual minority adolescents were more likely to experience homophobic bullying and illegal substance use compared with heterosexual individuals [40]. Another study proved the association between homophobic bullying victimization in school and substance abuse among LGBT adolescents [51]. Although in the present study, the association between homophobic traditional bullying and illegal substance use became insignificant after the multivariate logistic regression, the negative effects of homophobic traditional bullying on mental health and illegal substance use in sexual minority individuals still warrants monitoring and intervention. Research has found that anti-homophobic bullying policies, which are properly carried out, can significantly decrease the risk of alcohol and drug use among sexual minorities [52].

The present study found that homophobic cyberbullying victimization in childhood and adolescence is a powerful predictor of illegal substance use in early adulthood. Cyberbullying has different characteristics compared with traditional bullying, such as anonymity and individualistic activity [53]. Cyberbullies have lower neuroticism and higher agreeableness in comparison with traditional bullies [54]. For victims, a study of general adolescents demonstrated that compared with those who were only traditionally bullied, those who were cyberbullied were more likely to have depression, anxiety, and aggression [55]. The present study further supported the unique role of homophobic cyberbullying victimization in childhood and adolescence, in the use of illegal substances in early childhood. These results demonstrate the necessity of prevention, early detection, and intervention for homophobic cyberbullying among sexual minority individuals. Anti-bullying policies that are inclusive of sexual orientation [56] and Gay-Straight Alliances [57] are beneficial for LGBT individuals who suffer from traditional or cyber bullying. On the other hand, the significant association between homophobic bullying and illegal substance use can also be explained by MST [12]. Chronic stressors will accumulate and make subjects unable to tolerate and adopt, resulting in mental and substance use disorder [58]. In specific to sexual minorities, MST suggests that both distal (social) and proximal (psychological) stress are predominantly associated with poor health outcomes [38]. The result of the current study echoes the association between social stressors (homophobic bullying) and mental health concerns.

### 4.3. Disclosure of Sexual Orientation and Illegal Substance Use in Sexual Minorities

The present study found that disclosure of sexual orientation at junior high school was significantly associated with illegal substance use in early adulthood. Gay and bisexual adolescents at junior high school may lack effective coping strategies because of immature neurocognitive functions [59] and social skills [60]. Those individuals who come out in early adolescence may be less able to cope effectively with stressors related to the stigma of sexual minority identification, and they may struggle to deal with bullying incidents compared with those who reach sexual orientation milestones in their late adolescence or young adulthood. This can lead to negative mental health outcomes such as depression and anxiety [61,62]. Illegal substance use may be an ineffective strategy used by them to cope with stress. It is interesting to note that disclosure of sexual orientation at elementary school was not significantly associated with illegal substance use in early adulthood. The rate of participants in the present study who disclosed their sexual orientation at elementary school was less than 5%, which may limit the ability to draw conclusions on the relationship between disclosure of sexual orientation at elementary school and later illegal substance use. In addition, sexual orientation may not be the focus of attention among students at elementary school; therefore, sexual minorities may encounter severe difficulties in sexual orientation related adjustment that may increase their risk of later illegal substance use.

### 4.4. School Factors Associated with Illegal Substance Use in Sexual Minorities

Researchers have previously investigated the role of school performance and issues of mental health among sexual minorities [63]. The present study found that missing classes or truancy in senior high school was significantly associated with illegal substance use in early adulthood. For school children and adolescents, truancy was found to be significantly associated with substance use [64], and another study showed similar results [65]. However, no further studies have explored this association among sexual minorities, and the present study fills this gap in the literature. The univariate logistic regression analysis in the present study found that a low education level was significantly associated with illegal substance use in early adulthood, although the association became insignificant following multiple adjusted logistic regression analysis. A previous study reported that men with a lower educational level were at a higher risk of being hazardous drinkers and heavy cannabis users [66]. The cross-sectional research design of the present study limited the ability to determine the temporal causal relationship between a low education level and illegal substance use in gay and bisexual men. Further prospective studies are needed to examine whether both low education level and illegal substance use are the results of minority stress encountered by gay and bisexual men.

### 4.5. Family and Peer Support and Illegal Substance Use in Sexual Minorities

Family acceptance to sexual minorities predicts greater self-esteem and general health status among LGBT individuals [67]. More LGBT-supportive environments in school and communities may also predict less substance use for LGBT adolescents [68]. However, the present study did not find a significant association between family and peer support during childhood and adolescence, and illegal substance use in early adulthood. Although support from peers and family during childhood and adolescent was measured, the individuals current support was not measured, which may strongly contribute to current substance use.

### 4.6. Limitations

Several limitations of the current study should be addressed. First, as a cross-sectional study, it was not possible to determine causal relationships between homophobic bullying victimization and school factors in childhood and adolescence and illegal substance use in early adulthood. Second, the study data were exclusively self-reported. Therefore, the use of only a single data source could have influenced the findings and may have resulted in shared-method variances. Third, the study obtained data on participants’ homophobic bullying victimization, school factors, and family and peer support retrospectively, and therefore, recall bias might have been introduced. Finally, only gay and bisexual men were recruited into this study. Potential implications for lesbian or other sexual minorities is lacking. Therefore, the generalizability of this study is limited.

## 5. Conclusions

The present study revealed that homophobic cyberbullying victimization in childhood and adolescence, disclosure of sexual orientation at junior high school, and missing classes or truancy in senior high school were significantly associated with illegal substance use in early adulthood. Based on the results of the current study, the authors suggest that mental health professionals should routinely assess the experiences and impact of homophobic cyberbullying victimization in childhood and adolescence when approaching gay and bisexual men with illegal substance use. Timely referral to gay-affirmative cognitive behavioral therapy [69] for victims of homophobic bullying during their youth, could help prevent those using illegal substances to cope with these traumatic experiences. Moreover, the underlying reasons behind the significant association between disclosure of sexual orientation at junior high school and missing classes or truancy in senior high school with illegal substance use in early adulthood, warrants further investigation. These findings could then provide knowledge for the development of a prevention and early intervention strategy for illegal substance use in gay and bisexual men.

## Figures and Tables

**Table 1 ijerph-16-04476-t001:** Demographic factors and sexual orientation-related experiences associated with illegal substance use examined by univariate logistic regression (*N* = 500).

Variables	Mean	SD	B	OR	95% CI	*p*
Age (years)	22.94	1.57	0.13	1.14	0.86–1.51	0.370
	*n*	%	B	OR	95.0% of CI	*p*
Education level						
High (college or above)	450	90				
Low (high school or below)	50	10	1.31	3.70	1.38–9.94	**0.009**
Sexual orientation identity						
Gay	371	74.2				
Bisexual	129	25.8	−0.47	0.63	0.21–1.89	0.407
Time to disclose sexual orientation						
At elementary school						
No	476	95.2				
Yes	24	4.8	0.73	2.07	0.46–9.43	0.346
At junior high school						
No	365	73				
Yes	135	27	1.64	5.16	2.11–12.6	**<0.001**
At senior high school						
No	215	43				
Yes	285	57	0.98	2.66	0.97–7.34	0.058
At college or above						
No	53	10.6				
Yes	447	89.4	0.94	2.56	0.34–19.45	0.363
Victims of homophobic bullying						
Traditional bullying						
No	310	62				
Yes	190	38	1.10	3.00	1.24–7.32	**0.015**
Cyberbullying						
No	299	59.8				
Yes	201	40.2	1.44	4.22	1.62–10.98	**0.003**

CI = Confidence interval; OR = Odds ratio, ratio of odds of illegal substance use versus non-use among participants; SD = Standard deviation; bold values indicate statistical significance.

**Table 2 ijerph-16-04476-t002:** Family, peer, and school factors associated with illegal substance use examined by univariate logistic regression (*N* = 500).

Variables	Mean	SD	B	OR	95% CI	*p*
Perceived family support on the APGAR	8.49	3.83	−0.08	0.93	0.83–1.03	0.179
Perceived peer support on the APGAR	11.42	2.89	−0.02	0.98	0.85–1.13	0.807
	*n*	%	B	OR	95.0% CI	*p*
Parental marital status						
Married and living together ^a^	328	65.6	-	-	-	-
Separated or divorced	136	27.2	−0.11	0.90	0.34–2.35	0.830
Widowed	36	7.2	−18.23	<0.001	<0.01–<0.01	0.998
Paternal education level						
High (senior high school or above)	385	77				
Low (junior high school or below)	115	23	−0.02	0.98	0.36–2.72	0.975
Maternal education level						
High (senior high school or above)	388	77.6				
Low (junior high school or below)	112	22.4	0.28	1.32	0.51–3.45	0.576
Satisfaction with academic performance						
In elementary school						
High	401	80.2				
Low	99	19.8	0.18	1.20	0.43–3.34	0.725
In junior high school						
High	336	67.2				
Low	164	32.8	0.75	2.21	0.9–5.01	0.085
In senior high school						
High	310	62.0				
Low	190	38.0	0.32	1.38	0.58–3.26	0.463
Miss classes or truancy						
In elementary school						
No	456	91.2				
Yes	44	8.8	0.52	1.68	0.48–5.93	0.418
In junior high school						
No	414	82.8				
Yes	86	17.2	0.07	1.07	0.35–3.25	0.901
In senior high school						
No	359	71.8				
Yes	141	28.2	0.99	2.68	1.13–6.32	**0.025**

CI = Confidence interval; OR = Odds ratio; SD = Standard deviation; ^a^: reference; bold values indicate statistical significance; bold values indicate statistical significance.

**Table 3 ijerph-16-04476-t003:** Predictors of illegal substance use examined using forward stepwise logistical regression.

Variables	B	OR	95% CI	*p*
Disclosure of sexual orientation at junior high school	1.54	4.67	1.88–11.58	**0.001**
Miss classes or truancy in senior high school	0.93	2.52	1.04–6.13	**0.041**
Victims of homophobic cyberbullying	1.26	3.51	1.33–9.30	**0.011**

OR = Odds ratio; bold values indicate statistical significance.

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
