# Peer review of "Multi-Dimensional Factors Associated with Illegal Substance Use Among Gay and Bisexual Men in Taiwan"

_ijerph, 2019, doi:10.3390/ijerph16224476_

Round 1
Reviewer 1 Report
The results are very interesting, and show a reality of today's society in terms of sexual diversity and a serious problem regarding the consumption of illegal substances.
It is suggested to indicate and specify in the manuscript (Materials and Methods), the degree of spatial distribution that represents the results of the 500 participants in the survey. They represent a capital city, a large region or the entire nation ????.
The results of the univariate logistic regression analysis showed that victims of homophobic traditional bullying in school (OR = 3.00; P = 0.015) and victims of homophobic cyberbullying in school 214 (OR = 4.22; P = 0.003) were significantly associated with the use of illegal substances. On the Other hand, the results of the forward stepwise logistic regression revealed that victims of homophobic cyberbullying in school (OR = 1.26; P = 0.011), disclosure of sexual orientation at junior high school (OR = 4.67; P = 0.001), and missing classes or truancy in senior high school (OR = 2.52; P = 0.041) were significantly associated with the use of illegal substances. Considering that the majority of OR values are higher than 1.0, especially for demographic factors and sexual orientation, victims of traditional and cyberbullying, it would be very important to associate that socio-emotional factors are present in the consumption of illegal substances in this population (homosexual and bisexual).
It would be very interesting to associate the different forms of traditional bullying observed, with consumption of a specific illegal substances (forms of bullying v/s substance consumed).
There is additional information that could be included from seizures of them (Amounts of seizures) or from use of illegal substances (Quantitative) in Taiwan. Illegal substance trafficking. At least for marijuana, methamphetamine and cocaine.
It is suggested, to indicate an idea of the frequency of illicit substance use in this population, following the format used to determine the frequency of traditional bullyng and cyberbullying.
Author Response
Reply to Comments from the Reviewers:
Reviewer #1:
The results are very interesting, and show a reality of today's society in terms of sexual diversity and a serious problem regarding the consumption of illegal substances.
It is suggested to indicate and specify in the manuscript (Materials and Methods), the degree of spatial distribution that represents the results of the 500 participants in the survey. They represent a capital city, a large region or the entire nation ????.
Response: Thanks for your questions. This study recruited participants from on-line advertisements and printed ones at the LGBT student clubs at 25 colleges, where are all over in Taiwan. Recruited participants might come from anywhere in Taiwan; however, the questionnaires did not include residence. Therefore, we cannot provide the detailed distribution of participants, and we will add additional statements at line 120 as “…. centers for sexual minorities in Taiwan.” and line 121 as “…. student clubs at 25 colleges in Taiwan.”
The results of the univariate logistic regression analysis showed that victims of homophobic traditional bullying in school (OR = 3.00; P = 0.015) and victims of homophobic cyberbullying in school 214 (OR = 4.22; P = 0.003) were significantly associated with the use of illegal substances. On the Other hand, the results of the forward stepwise logistic regression revealed that victims of homophobic cyberbullying in school (OR = 1.26; P = 0.011), disclosure of sexual orientation at junior high school (OR = 4.67; P = 0.001), and missing classes or truancy in senior high school (OR = 2.52; P = 0.041) were significantly associated with the use of illegal substances. Considering that the majority of OR values are higher than 1.0, especially for demographic factors and sexual orientation, victims of traditional and cyberbullying, it would be very important to associate that socio-emotional factors are present in the consumption of illegal substances in this population (homosexual and bisexual).
Response: Thanks for your reply, and the associated factors are listed in Table 1.
It would be very interesting to associate the different forms of traditional bullying observed, with consumption of a specific illegal substances (forms of bullying v/s substance consumed).
Response: Thanks for your questions. We categorize traditional bullying into social bullying, verbal bullying, physical bullying, and snatching of belongings. Moreover, we use univariate logistic regression to identify the association between all kinds of illegal substance and four kinds of traditional bullying. We will add some descriptions in the line 149~152 as “…traditional bullying. In order to group traditional bullying, we categorized social exclusion into “social bullying”, name calling along with ill speaking into “verbal bullying”, physical along with forced work into “physical abuse”, and confiscation of money/supplies into “snatching of belongings”.”, and in the line 237~241 as “Moreover, the association between subtypes of traditional bullying and specific illegal substance were also identified by univariate logistic regression. We found that social bullying was significantly associated with use of methamphetamine (OR=5.52; P=0.028). On the other hand, there were insignificantly trend of association between use of marijuana versus snatching of belongings (OR=10.00; P=0.051), and use of methamphetamine versus verbal bullying (OR=4.79; P=0.063).”
It is suggested, to indicate an idea of the frequency of illicit substance use in this population, following the format used to determine the frequency of traditional bullying and cyberbullying.
Response: Thanks for your questions. We will add additional sentences to describe frequency of illegal substance and bullying in line 213~221 as” The frequency of illegal substance use are as following: marijuana (never used: 99%; ever used but less than once per month: 1%), methamphetamine (never used: 97%; ever used but less than once per month: 1.8%; once per month but less than twice per month: 0.2%; twice to four times per month: 1%), ketamine (never used: 98.6%; ever used but less than once per month: 1.2%; once per month but less than twice per month: 0.2%), ecstasy (never used: 98.8%; ever used but less than once per month: 1%; once per month but less than twice per month: 0.2%), volatile organic compounds (never used: 99.6%; ever used but less than once per month: 0.2%; twice to thrice per week: 0.2%). The frequency of bullying are as following: traditional bullying (never: 13%; just a little: 49%; often/all the time: 38%) and cyberbullying (never: 59.8%; just a little: 32.6%; often/all the time: 7.6%).”
There is additional information that could be included from seizures of them (Amounts of seizures) or from use of illegal substances (Quantitative) in Taiwan. Illegal substance trafficking. At least for marijuana, methamphetamine and cocaine.
Response: Thanks for your questions. We will review related articles about use of illegal substance in Taiwan. On the other hand, cocaine is not popular in Taiwan. Several will be added into manuscript in line 47~52 as “In Taiwan, a cross-sectional study indicated 16% of recreational drug use in the previous 6 months within men who have sex with men (MSM) [Ko NY et al. 2012], and it is also higher than an epidemiological study, which reported 0.17% of past-1-year prevalence for club drug use in general population [Chen WJ et al. 2017]. To be specific, methamphetamine along with ketamine and marijuana are popular according to a national survey [Chen WJ et al. 2017]; however, such investigation for sexual minorities is insufficient.”
Reviewer 2 Report
This manuscript intends to identify factors associated with illicit substance use in gay and bisexual men in Taiwan. This is an important and noteworthy topic as substance use behaviours and associated factors may vary because of national policies, laws and belief systems. More research on this topic in non-Eurocentric countries and cultures is, therefore, more than welcome. The manuscript is well-written and of high quality. There are some minor issues as outlined below:
Abstract: A lot of effort is put into describing the data collection methods at the expense of analysis and discussion. A stronger focus on results and discussion would be beneficial.
Introduction (general): There is a lack of data on substance use within this population as well as a lack of comparisons. A general comparison to the general population or those with a majority sexual identity, particularly from Taiwanese studies, would assist in the interpretation of substance use disparities.
A factor that might have relevance here as well, particularly considering the authors’ findings regarding the association between illicit substance use and disclosure, is that this critical developmental period coincides with the typical age of onset of alcohol and tobacco use, as well as illicit drug use in adolescents and young adults.
p. 2, 51-55: A stronger use of MST would be helpful; particularly Meyer’s contextualisation of distal and proximal stressors are helpful in categorising and contextualising the various factors that may influence substance use in this population. It would also be beneficial to discuss this stronger in the discussion.
p. 3, 115-117: The authors write “those who exhibited any deficits (e.g. substance use or intellectual disability) were not included in this study”. The term ‘deficits’ is unclear in this context and needs to be defined, particularly considering that this study looks at substance use.
p. 3, 123-124: please comment on the validity and reliability of the DUDIT-E. You may want to provide more information on this tool in general.
p. 3, 124-125: please justify the inclusion of these substances above others.
p. 5, 188-195: there is some awkward language in this paragraph. Consider rewriting.
p. 9, 313-315: this is an important limitation considering that research is extremely scarce in this population, which leads me to the question: why this approach has been taken? Please also comment on the generalisability of these results for gay and bisexual men.
Author Response
Reply to Comments from the Reviewers:
Reviewer #2:
This manuscript intends to identify factors associated with illicit substance use in gay and bisexual men in Taiwan. This is an important and noteworthy topic as substance use behaviours and associated factors may vary because of national policies, laws and belief systems. More research on this topic in non-Eurocentric countries and cultures is, therefore, more than welcome. The manuscript is well-written and of high quality. There are some minor issues as outlined below:
Abstract: A lot of effort is put into describing the data collection methods at the expense of analysis and discussion. A stronger focus on results and discussion would be beneficial.
Response: Thanks for your suggestions. We rewrite the abstract as following: “Illegal substance use in sexual minorities is an important health issue worldwide. The present cross-sectional study aimed to investigate the multi-dimensional factors associated with illegal substance use among gay and bisexual men in Taiwan. This questionnaire-survey study recruited 500 gay or bisexual men aged between 20 and 25 years. Their experiences of using eight kinds of illegal substance in the preceding month were collected. Their previous experiences of homophobic bullying, satisfaction with academic performance, truancy, perceived family and peer support in childhood and adolescence, and social-demographic characteristics, were also collected. Potential factors associated with illegal substance use were identified using univariate logistic regression, and further selected into a forward stepwise logistic regression model to identify the factors most significantly related to illegal substance use. A total of 22 (4.4%) participants reported illegal substance use in the preceding month, and mean age was 22.9±1.6. Forward stepwise logistic regression revealed that being victims of homophobic cyberbullying in childhood and adolescence (odds ratio [OR] = 1.26; P = 0.011), disclosure of sexual orientation at junior high school (OR = 4.67; P = 0.001), and missing classes or truancy in senior high school (OR = 2.52; P = 0.041) were significantly associated with illegal substance use in early adulthood. Multi-dimensional factors in childhood and adolescence that were significantly associated with illegal substance use in early adulthood among gay and bisexual men were identified. Besides traditional bullying, the effect of cyberbullying and school performance on illegal substance use should not be ignored. This study is limited to the cross-sectional design and possible recall bias. Mental health professionals must routinely assess these significant factors to prevent and intervene in illegal substance use among gay and bisexual men.”
Introduction (general): There is a lack of data on substance use within this population as well as a lack of comparisons. A general comparison to the general population or those with a majority sexual identity, particularly from Taiwanese studies, would assist in the interpretation of substance use disparities.
Response: Thanks for your suggestions, and we will add some sentences in line 47~52 as “In Taiwan, a cross-sectional study indicated 16% of recreational drug use in the previous 6 months within men who have sex with men (MSM) [Ko NY et al. 2012], and it is also higher than an epidemiological study, which reported 0.17% of past-1-year prevalence for club drug use in general population [Chen WJ et al. 2017]. To be specific, methamphetamine along with ketamine and marijuana are popular according to a national survey [Chen WJ et al. 2017]; however, such investigation for sexual minorities is insufficient.”
A factor that might have relevance here as well, particularly considering the authors’ findings regarding the association between illicit substance use and disclosure, is that this critical developmental period coincides with the typical age of onset of alcohol and tobacco use, as well as illicit drug use in adolescents and young adults.
Response: Thanks for your questions. We do have data about alcohol use and smoking among the 500 gay/bisexual men; however, due to insufficient data for dose of alcohol consumption per time, we cannot set the proper cutoff point for “problematic alcohol use” and “social drinking”. We only have the frequency of alcohol use. Then we cannot make further analysis to estimate association between risk factors and alcohol use. So does it for the data of cigarettes consumption.
P2, 51-55: A stronger use of MST would be helpful; particularly Meyer’s contextualization of distal and proximal stressors are helpful in categorizing and contextualizing the various factors that may influence substance use in this population. It would also be beneficial to discuss this stronger in the discussion.
Response: Thanks for your suggestions, and we will add some sentences in line 288~294 as “On the other hand, the significant association between homophobic bullying and illegal substance use can also be explained by MST [Meyer 1995]. Chronic stressors will accumulate and make subjects unable to tolerate and adopt, resulting in mental and substance use disorder.[Brady KT et al. 2005]. In specific to sexual minorities, MST suggests that both distal (social) and proximal (psychological) stress are predominantly associated with poor health outcomes. [Meyer et al. 2003] The result of current study echo the association between social stressors (homophobic bullying) and mental health concerns.”
P3, 115-117: The authors write “those who exhibited any deficits (e.g. substance use or intellectual disability) were not included in this study”. The term ‘deficits’ is unclear in this context and needs to be defined, particularly considering that this study looks at substance use.
Response: Thanks for your questions, and we will modify the sentence into “Those who exhibited any cognitive impairment (e.g. substance intoxication or intellectual disability) that………”
P3, 123-124: please comment on the validity and reliability of the DUDIT-E. You may want to provide more information on this tool in general.
Response: Thanks for your questions, and we will add some sentences in line 129~132 as “The D-score of Drug Use Disorders Identification Test-Extended (DUDIT-E) was used to identify the history of illegal substance use for all individuals. It had been developed for sequential clinical assessment of drug use. The concurrent validity of D-score is reported to be acceptable, and test-retest reliability is 0.79, indicating an excellent intraclass correlation [Berman, A. et al. 2007].”
P3, 124-125: please justify the inclusion of these substances above others.
Response: Thanks for your questions. We identify illegal substance use according to self-reported DUDIT-E. The data of other substance (eg. nicotine, alcohol…) is not analyzed because we focus on “entirely illegal” substance in Taiwan for this manuscript.
P5, 188-195: there is some awkward language in this paragraph. Consider rewriting.
Response: Thanks for your suggestions, and we will rewrite this paragraph as following “Initially, we summarized the demographic, sexual orientation, family, school and social support characteristics in Table 1. In order to estimate the odds ratio (OR) for multiple variables, univariate logistic regression was used to identify potential predictors associated with illegal substance use. Second, all potential predictive variables (P<0.05) identified from the first step were selected in a forward stepwise logistic regression model to determine the best predictors for illegal substance use. All tests were 2-tailed, and statistical significance was set at P < 0.05. All data were processed using SPSS version 23.0 for Windows (SPSS Inc., Chicago, IL).”
P9, 313-315: this is an important limitation considering that research is extremely scarce in this population, which leads me to the question: why this approach has been taken? Please also comment on the generalisability of these results for gay and bisexual men.
Response: Thanks for your questions. This study derived from a proposal to estimate the association between multiple factors and mental health outcome, including substance use. This proposal recruited only gay and bisexual men, and studies for other sexual minorities are still ongoing. Then we will rewrite the limitation as “…Finally, only gay and bisexual men were recruited into this study. Potential implications for lesbian or other sexual minorities is lacking. Therefore, the generalizability of this study is limited. ”